# Resolution of physics and deep learning-based protein engineering filters: A case study with a lipase for industrial substrate hydrolysis

Spencer Gardiner[1], Peter Dollinger[2], Filip Kovacic[2,3], Jörg Pietruszka[4,5], Daniel H. Ess[6], Karl-Erich Jaeger[2,7], Gunnar F. Schröder[8,9], Dennis Della Corte[1]*

1 Department of Physics and Astronomy, Brigham Young University, Provo, Utah, United States of America, 2 Institute of Molecular Enzyme Technology, Heinrich Heine University Düsseldorf, Jülich, Germany, 3 Department of Surgery, Massachusetts General Hospital, and Harvard Medical School, Boston, United States of America, 4 Institute of Bio- and Geosciences, Biotechnology (IBG-1), Forschungszentrum Jülich, Jülich, Germany, 5 Institute of Bioorganic Chemistry, Heinrich Heine University Düsseldorf Located at Forschungszentrum Jülich, Jülich, Germany, 6 Department of Chemistry and Biochemistry, Brigham Young University, Provo, Utah, United States of America, 7 Institute of Bio- and Geosciences IBG-1: Biotechnology, Forschungszentrum Jülich GmbH, Jülich, Germany, 8 Institute of Complex Systems, Structural Biochemistry (ICS-6), Forschungszentrum Jülich, Jülich, Germany, 9 Physics Department, Heinrich-Heine-Universität Düsseldorf, Düsseldorf, Germany

* dennis.dellacorte@byu.edu

## Abstract

Computational enzyme design remains a powerful yet imperfect tool for optimizing biocatalysts, especially when targeting non-natural substrates. Using design tools we investigated *Pseudomonas aeruginosa* LipA, a lipase with a flexible lid domain crucial for substrate binding and turnover, aiming to enhance its hydrolysis of the industrially relevant substrate Roche ester. We generated an initial set of single-point mutations based on structural proximity to the active site and evaluated their effects using a computational pipeline integrating molecular dynamics (MD) simulations, density functional theory (DFT) calculations, and ensemble-based energy scoring. While we identified several active variants, attempts to rank them by activity using structural features, such as hydrogen bond formation or residue flexibility, failed. Deep learning models, applied *post hoc* for structural analysis via AlphaFold3, produced nearly identical active site geometries across variants, irrespective of activity. Reaction pathway analysis revealed energy barriers varying by 5–15 kcal/mol depending on substrate conformation, with the nucleophile addition step consistently rate-limiting. However, these small energetic shifts, likely critical for incremental activity changes, were indistinguishable by current computational or deep learning methods. Our results highlight the limitations of existing approaches in resolving subtle functional differences and underscore the need for improved benchmarks, reactive force fields, and more sensitive ranking metrics. Advancing these areas will be essential for designing enzymes with gradual, evolution-like activity improvements and for bridging the gap between structural prediction and catalytic function.

**Data availability statement:** All experimental relevant data are within the manuscript and its Supporting Information files. All simulation relevant data (.pdb, .xtc, PLACER outputs) are freely available at https://simtk.org/projects/lipadesign.

**Funding:** D.DC and S.G. were supported by the National Institute of General Medical Sciences of the National Institutes of Health under award number R15GM155803. D.DC. and P.D. thank the graduate school iGRASPseed for funding. Part of this study was supported by the Deutsche Forschungsgemeinschaft (DFG, German Research Foundation) through funding no. JA 448/8-1 to KEJ. G.F.S. and D.DC. The funders had no role in study design, data collection and analysis, decision to publish, or preparation of the manuscript.

**Competing interests:** The authors have declared that no competing interests exist.

## Introduction

Enzymes are extraordinary catalysts, capable of accelerating reactions by many orders of magnitude with remarkable selectivity and efficiency [1]. These characteristics make enzymes attractive for sustainable manufacturing and green chemistry applications [2,3]. However, enzymes are rarely suitable for direct use in industrial processes without modification, as evolution has optimized them for cellular contexts rather than non-natural environments [4]. Consequently, enzymes often require extensive optimization to function effectively under non-physiological conditions or to catalyze novel chemical transformations [5]. This inherent limitation has driven interest in enzyme engineering, with the goal of designing bespoke biocatalysts for specific applications [6].

Despite progress in enzyme engineering, generating highly efficient catalysts remains a formidable challenge [1]. Directed evolution has been a cornerstone of enzyme optimization, yet it comes with significant limitations: it is resource-intensive, time-consuming, and often constrained by the availability of starting points for desired reactions [7]. While ultrahigh-throughput screening and continuous evolution platforms have expanded the potential of directed evolution, these methods still struggle to address the vast diversity of chemical transformations [1]. Enzyme promiscuity offers some solutions, providing a foundation for evolving new functions, but a generalizable strategy for designing efficient enzymes from scratch remains elusive [8].

*De novo* enzyme design has emerged as a promising strategy, leveraging computational tools to construct novel catalysts from first principles [9]. This approach can incorporate both natural and unnatural amino acids, as well as metal cofactors, to achieve desired reactivity. Computational methods often involve designing theozymes and selecting or hallucinating protein scaffolds, followed by packing optimization to refine active site geometry [10–13]. While this dramatically expands the design space, achieving efficient enzymatic catalysis requires an extraordinary degree of precision, as even minor deviations in side-chain positioning can drastically impair function [1]. The active site must stabilize the transition state while discriminating against the ground state, a delicate balance that often hinges on angstrom-level accuracy. Many enzymatic reactions involve multiple transition states and require orchestrated conformational changes to progress through each catalytic step [14]. Structural analyses have shown that scaffold-based designs often misplace polar side chains and fail to generate the necessary hydrogen-bonding networks, limiting their catalytic potential [1]. Techniques like explicit solvent molecular dynamics (MD) simulations and AI-driven structure prediction with design constraints have been proposed as promising strategies to overcome these limitations, but their effectiveness remains to be fully evaluated [15]. An alternative compromise involves computationally redesigning existing proteins, where subtle modifications to an active site can shift substrate selectivity.

Computational techniques have become indispensable for investigating protein-ligand interactions [16], with recent AI advancements accelerating progress in enzyme engineering. Molecular docking [17,18], MD simulations, and experimental

validation collectively enable the creation of novel biosensors [19] and unravel complex protein mechanics [20]. Deep learning models, inspired by natural language processing, enhance our ability to predict protein contacts [21–23], structures [24, 25], and complex formations [26]. Tools like BayesDesign [27], RFDiffusion [28], and ProteinMPNN [29] offer promising avenues for bridging the gap between theoretical designs and functional enzymes [28,30]. Simultaneously, MD simulations can reveal differences in conformational ensembles of enzyme variants, while density functional theory (DFT) calculations provide insights into transition state energetics. Analyzing hydrogen-bonding networks during the catalytic cycle highlights critical interactions, and tools like AlphaFold3 [26] and PLACER [15] can predict mutant-substrate complexes and cluster conformational ensembles along the reaction coordinate. Despite these advances, existing models often fail to accurately rank small iterative improvements in catalytic activity, emphasizing the need for integrated, physics-informed approaches that capture the full complexity of enzyme dynamics and catalysis.

Serine hydrolases present a challenging design problem, as their catalytic mechanism involves complex residue coordination and intricate hydrogen bond networks [15,28,31,32]. The catalytic cycle of serine hydrolases involves four steps (Fig 1) [33]. The substrate binding to the apoenzyme, where the catalytic serine—deprotonated by a histidine—attacks the carbonyl carbon of the ester, forming the first tetrahedral intermediate (TI1). Next, the histidine protonates the leaving group, facilitating its departure and generating the covalently linked acyl-enzyme intermediate (AEI). A water molecule, activated by the histidine, then attacks the AEI to form a second tetrahedral intermediate (TI2), which finally collapses upon proton transfer to release the product and restore the free enzyme. State-of-the-art design tools struggle to scaffold the required residue geometry with sufficient accuracy, and destabilizing the AEI as needed adds another layer of complexity. Overcoming these challenges requires new methods to assess structural compatibility with each step of the reaction, combining high-precision computational modeling with iterative experimental validation [34,35].

Lipases produced by bacteria of the genus *Pseudomonas* have been gaining much attention, for basic research as well as for industrial applications [36–39]. The lipase LipA [40] from *Pseudomonas aeruginosa* (shown in Fig 2A) is the first lipase of subfamily I.1 [41] with solved crystal structure [42] and has therefore often been used as a model lipase. It shows an extraordinary catalytic reactivity [40], high stability in organic solvents and non-aqueous media. Its use for the production of biodiesel and other industrially relevant esterification and transesterification reactions have also been shown [43,44]. Furthermore, LipA was successfully engineered for altered substrate specificity and enantioselectivity [45–47]. Moreover, LipA serves as a model for the homologous lipase from *Burkholderia glumae,* which is used as a biocatalyst for the kinetic resolution of racemic alcohols and amines in industrial processes operated at a ton scale by the world's largest chemical company BASF AG [48].

The Roche ester, (R)-methyl-3-hydroxy-2-methylpropionate, is a crucial chiral building block in the synthesis of pharmaceuticals, vitamins, fragrance components, antibiotics, and other valuable natural products [49,50]. Despite its industrial importance, few biocatalytic routes for Roche ester synthesis have been reported, and existing enzymatic processes often suffer from low efficiency [50]. Designing an enzyme capable of effectively hydrolyzing Roche ester could unlock more sustainable and cost-effective production pathways for a variety of high-value compounds. This potential makes Roche ester an ideal target for enzyme engineering efforts, providing a clear benchmark for evaluating the success of computational and experimental redesign strategies.

**Fig 1. A simplified mechanism of LipA hydrolyzing the Roche Ester.**

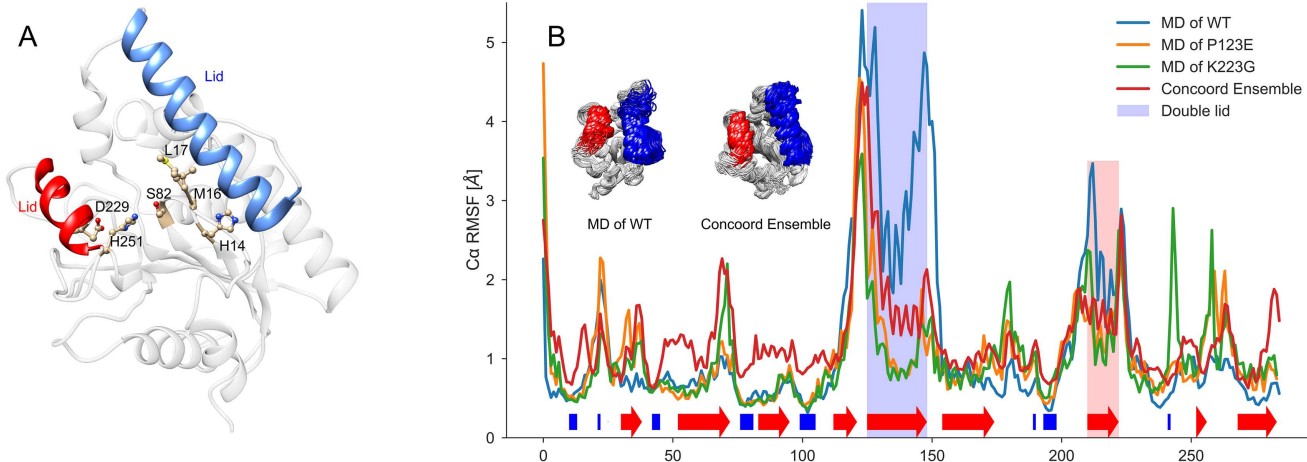

**Fig 2. Dynamics of LipA.** *A) The crystal structure of LipA (PDB ID 1EX9) is shown, highlighting the double lid α-helices (red and blue), as well as residues of high importance (catalytic triad S82, H251, D229 and mutational sites H14, M16, L17). B) LipA MD results: The root mean square fluctuation of Cα-atoms of LipA and variants P123E and K223G. Blue and red shaded regions represent the α-helical double lid. Inserted are structure images of 96 frames from the wild-type MD-simulation and the CONCOORD ensemble (residues corresponding to the double lid highlighted in red and blue).*

In this study, we tested integrative physics-based simulations with an experimental iterative approach to enhance activity of LipA with Roche ester, a very poor substrate of LipA. We generated a library of single- and double-point LipA variants, tested their activities, and evaluated the stability of our selection algorithm alongside deep learning methods for variant ranking. Our results showed that existing computational ranking methods were highly sensitive to substrate parameterization, often missing subtle activity changes. To address this, we adopted a multi-stage approach: performing detailed DFT calculations of transition states, conducting molecular dynamics simulations to assess transition state visitation rates, and using AI models like AlphaFold3 and PLACER for structural clustering.

## Results

### Molecular dynamic simulations can elucidate the impact of some mutations on LipA dynamics

To assess how specific mutations could affect the structural dynamics of LipA, critical for enzyme function, we first analyzed the conformational fluctuations of wild-type LipA, by using fully solvated 100 ns classical molecular dynamics (MD) simulation conducted with GROMACS (Fig 2A). Notably, the fluctuations of two α-helices covering the active site residues located within a hydrophobic cleft (Fig 2B) were among the most pronounced. This flexible α-helical domain, known as the double lid, undergoes an open-closed transition to permit substrate access to the active site [42,51].

Next, we mutated the residues P123E and K223G at the hinge regions of the double lid, followed by 100 ns MD simulations to examine potential perturbations in native fluctuations. Given that mutated residues are in surface-exposed flexible loops that generally tolerate mutations without perturbing protein structure (Fig 2a), LipA unfolding was deemed unlikely, suggesting a preserved overall structure. The simulations show that the double lid in wild-type LipA exhibited substantially stronger fluctuations – up to 3 Å RMSF (Fig 2b) – compared to the mutated variants, indicating that the mutations could strongly influence protein function through altered double lid dynamics.

Therefore, to assess the functional impact of the P123E and K223G mutations, single-conformation energy calculations are insufficient. Instead, a representative ensemble of LipA conformations, capturing both open and closed lid states, is necessary to accurately evaluate the mutations' effects.

## Conformational ensembles can be generated with physics-based models

To generate an ensemble of protein structures [52] by sampling conformations within structure-based geometric restraints we have used the CONCOORD program [53]. Unlike MD simulations, where the ensemble width depends on simulation time and the occurrence of rare events [54], CONCOORD provides a reproducible ensemble with a well-defined root mean square deviation (RMSD) around the starting conformation. This reproducibility is particularly advantageous for capturing relevant structural diversity without the unpredictability of rare conformational shifts that may not represent the native equilibrium ensemble [55]. Careful consideration was given to modeling the transition state [ES*], involving the catalytic triad of LipA—residues S82, D229, and H251 [42]. The LipA crystal structure (Fig 2a), co-crystallized with the substrate analog 1,2-dioctylcarbamoyl-glycero-3-O-p-nitrophenyl octylphosphonate, captures the active site in a transition state geometry, which was preserved across all designed structures.

The crystal structure was chosen as the starting point for CONCOORD rather than an equilibrated MD frame to ensure the tetrahedral conformation of the co-crystallized substrate was maintained throughout the ensemble. Analysis of the MD trajectory revealed an average root mean square fluctuation (RMSF) of 2.5 Å from the crystal structure over 100 ns, indicating that an arbitrary MD frame could introduce significant deviations from the ideal transition state geometry. To mitigate this, additional constraints were applied in CONCOORD to anchor the catalytic triad residues (Fig 2a) in their transition state configuration while allowing the remaining residues to sample structural variability. This approach produced an ensemble with a defined structural width, while preserving an active site geometry identical to the crystal structure. As shown in Fig 2b, the 95 CONCOORD-generated wild-type LipA structures recapitulate the conformational ensemble observed in the 100 ns MD trajectory, though with reduced fluctuations of the double lid motif due to the imposed transition state constraints favoring an open conformation. For this study, a final ensemble of 96 structures was assembled, comprising the 95 CONCOORD structures and the original crystal structure.

## Predicted single- and double-point mutants show increase in LipA activity towards Roche ester

We aimed predicting mutations that can increase the low enzymatic activity of the wild type LipA ($0.062 \pm 0.004$ $U$/mg) with Roche ester as determined by GC analysis after incubation of the enzyme with Roche ester for 48 h. We used an ensemble-based simulation protocol (Supplementary Information S6) to select promising single point candidate mutations in presence of Roche ester at residues H14, M16, L17, and H81, all located near the active site on flexible loop regions (Fig 2A). These four variants were selected from 669 computationally viable mutations based on predicted transition state stabilization ($\Delta\tilde{E} < 0.985$), evolutionary conservation (PSSM > −4.0), and proximity to substrate (<5 Å). These positions were unlikely to negatively impact protein folding or overall function. The three highest-ranked single-point mutations, along with two lower-scoring variants (M16T and H81S), were selected for experimental validation. These variants were produced, purified (S3–S5 Figs), and activated *in vitro* as described previously [56]. Compared to wild-type LipA, all five variants exhibited increased activity towards the Roche ester substrate, with activity enhancements ranging from ~230% to ~930% (Table 1). The H14G mutation, ranked as the most promising variant, yielded the highest activity increase (~930%), whereas the lowest-ranked variant, H81S, achieved a more modest improvement (~230%). Interestingly, while these mutations enhanced Roche ester hydrolysis, they significantly reduced activity towards the bulkier, aromatic *p*-NPB substrate commonly used to assay LipA activity (Table 1). These results highlight the specificity of the computational protocol, which effectively optimized LipA for Roche ester hydrolysis at the expense of broader substrate compatibility.

Building on the single-point variant results, double-point mutants were explored to further enhance LipA activity [57]. Using H14G and M16A—the most promising single-point variants—as templates, a total of 3090 double mutants were generated. Of these, 2669 mutants were successfully evaluated using ensemble-based activation energy calculations (Supplementary Information S6). Three low-energy variants within 5 Å of the active site were selected for experimental validation, alongside two low-energy variants independent of distance to the active site. As negative controls showing high energies, M16A_I142Y and M16A_G139F were also included (Table 1).

**Table 1. The hydrolytic activity of wild type and LipA variants towards Roche ester and para-nitrophenyl butyrate (p-NPB). Activity with Roche ester was determined by GC quantification of substrate after incubating an enzyme and substrate for 48h. Lipase activity with pNPB substrate was determined spectrophotometrically. Activities are expressed as a percentage, normalized against the activity of the wild-type (WT) enzyme.**

| Mutant | Roche ester Activity (%) | p-NPB Activity (%) |
|---|---|---|
| H14G | 932.4 | 20.6 |
| M16T | 603.4 | 19.0 |
| L17G | 561 | 41.4 |
| M16A | 476.5 | 67.3 |
| H14G_R56N | 411.1 | 11.4 |
| H14G_L17A | 262.4 | 0.9 |
| H81S | 233.4 | 1.0 |
| M16A_F214I | 187.2 | 9.2 |
| WT | 100 | 100 |
| M16A_G139F | 0 | 2.7 |
| M16A_H14P | 0 | 0.7 |
| M16A_R56K | 0 | Na |
| M16A_I142M | 0 | Na |
| M16A_I142Y | 0 | 40.5 |

These eight double-point variants were produced, purified, and activated simultaneously with single-point variants to ensure experimental consistency. The results showed that while double variants carrying the H14G mutation remained active, their activity dropped below ~410% compared to wild-type, significantly lower than the single H14G mutation (~930%). In the case of M16A-based variants, five out of six variants showed no Roche ester hydrolysis and reduced activity with p-NPB. The only one M16A-series double variant, M16A_F214I, exhibited increased (~180%) activity compared to wild-type LipA. These findings suggest that the M16A mutation introduces structural constraints that are incompatible with additional mutations, therefore it cannot be used for enhancing Roche ester hydrolysis. A similar phenomenon has been observed in the *P. aeruginosa* TE3285 lipase, where M16 mutations impaired amidase activity, despite this enzyme differing from LipA by only one residue (V156I) [58].

### Rank order of mutations is highly dependent on substrate parametrizations and selected frames

The ranking of mutations is highly sensitive to substrate parametrizations and the chosen frames. We re-evaluated the mutational candidates using various ensemble frames and parametrization schemes. The resulting mutation rankings across different ensembles and parametrizations were inconsistent. Therefore, no conclusions can be drawn regarding the ability of the ensemble-based design strategy to consistently predict the active LipA variants from applied ranking.

### Detailed DFT analysis of the Roche ester hydrolysis reaction within LipA

M06-2X density functional theory (DFT) calculations of an active site model were conducted to obtain transition-state structure models, which were used to inform bond length constraints in MD simulations that modelled said transition states. Serine hydrolases typically follow a double-displacement mechanism, where a serine nucleophile forms an acyl-enzyme intermediate (labelled Acyl Intermediate in Fig 3) that is then hydrolyzed by an activated water molecule. For this work we used a model where four distinct transition states were identified: ester addition (TS 1), methanol elimination (TS 2), hydroxide addition (TS 3), and acid elimination (TS 4), as well as two tetrahedral intermediates (TI 1/TI 2 – Fig 3). In all viable transition states, four functionally relevant hydrogen bonds shown in Fig 4 (labeled 1–4) were maintained.

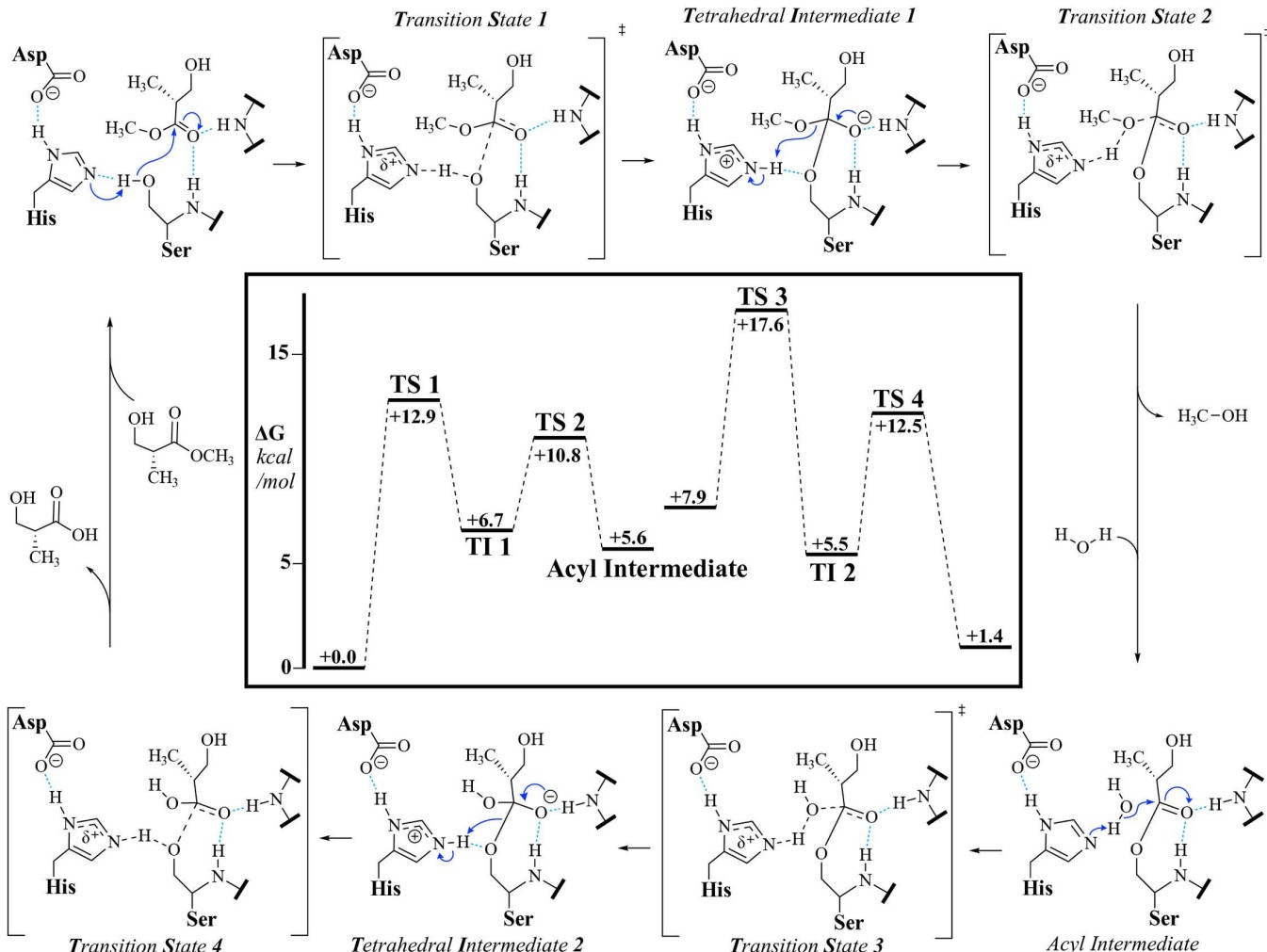

**Fig 3. The reaction mechanism of LipA hydrolyzing the (R)-Roche ester and a model reaction coordinate for the active site theozyme.** ΔG values shown in the reaction coordinate are for a "linear" substrate conformation, where the tail hydroxyl group does not hydrogen bond with the carbonyl oxygen or the ester oxygen.

Fig 3 shows stepwise energy barriers ranging from 5 to 15 kcal/mol, with transition states 1 and 3 consistently presenting the highest barriers. Notably, the ester's chirality had no significant effect on the energy landscape.

Conformation explorations performed with CREST found no substantial impacts on energy barriers due protein side chain conformations, however, substrate conformation significantly influenced the energy barriers of each step. The largest energy decrease (5 kcal/mol) occurred when the substrate's hydroxyl group formed a hydrogen bond with either the ester or carbonyl oxygen.

### Protein dynamics and hydrogen network analysis does not differentiate between active and inactive LipA variants

No correlation was observed between hydrogen bond formation frequency and enzymatic activity (Fig 5). While the occurrence of specific hydrogen bonds varied significantly depending on their location (e.g., H-bond 1 vs. H-bond 3, Fig 5), these frequencies remained unchanged across mutants with diminishing activity. A binary classification of mutants into active and

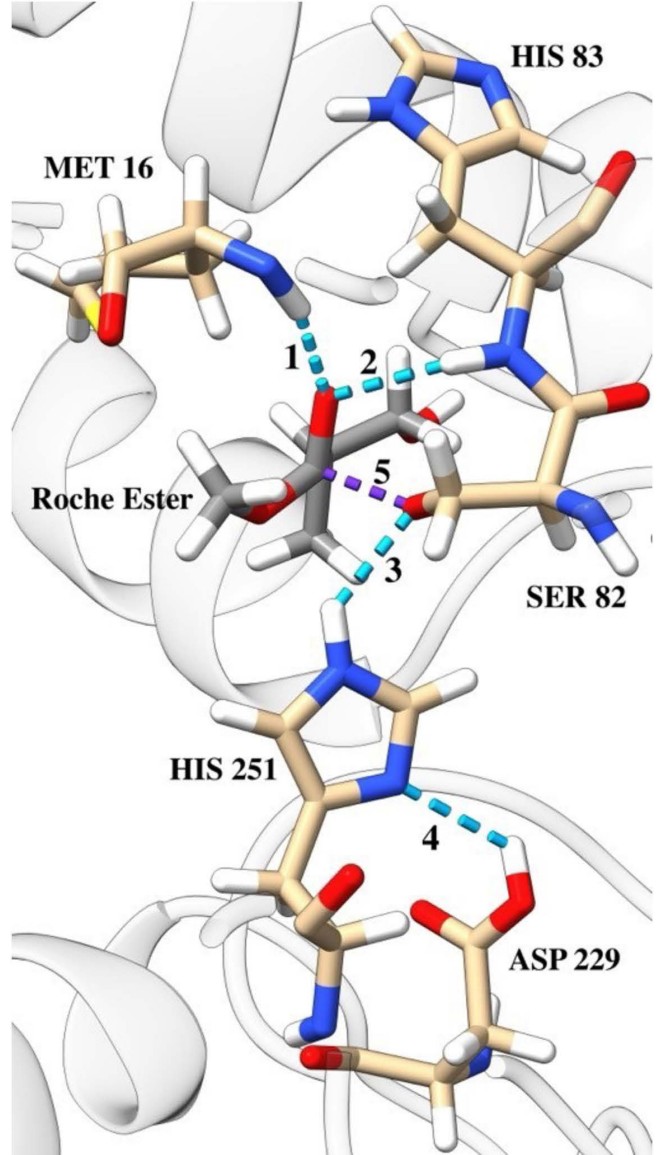

**Fig 4. The active site of the LipA during transition state 1 (TS 1).** *The purple dashed line (labeled 5) represents the covalent bond that forms between the ester and serine 82. The light blue lines labeled 1 through 4 represent key hydrogen bonds that facilitate and stabilize the hydrolysis reaction.*

inactive groups similarly showed no significant differences in hydrogen bond formation count, suggesting that catalytic efficiency is not directly tied to the persistence of these interactions. Hydrogen bonds were identified using standard geometric criteria (donor-acceptor distance ≤ 3.0 Å, donor-hydrogen-acceptor angle ≥ 150°) as implemented in the MDAnalysis package, and alternative distance thresholds (2.5–4.0 Å) produced similar random distributions without revealing activity-dependent patterns.

To further investigate structural dynamics, we analyzed the RMSF of residues in MD simulations. Mutants generally exhibited higher RMSF values than the wild-type enzyme, indicating increased residue mobility. However, this heightened flexibility did not correlate with enzymatic activity, as both active and inactive variants showed comparable fluctuation

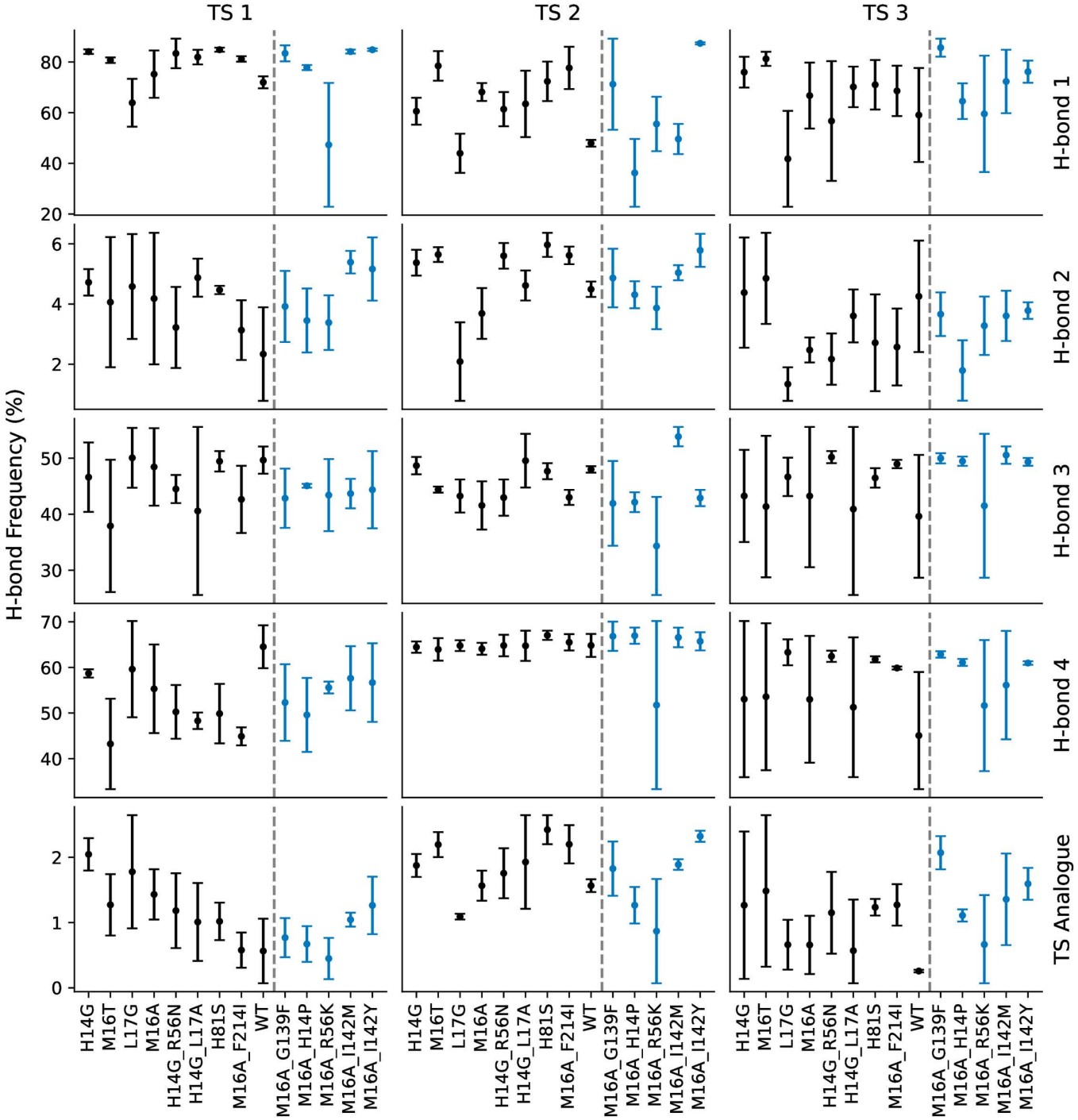

**Fig 5. The percentage frequency of hydrogen bond formation in MD simulations of the reaction steps.** *Each plot shows the average frequency of H-bond formation as a percent of the total simulation time (black and blue dots). The X-axis labels are the enzyme variants, sorted by activity high to low, left to right. Variants to the right of the dotted line (blue dots) showed 0 experimental activity towards the Roche ester. The labels to the right of the rows correspond to the H-bonds shown in Fig 3. Last row (Transition State analogue) shows the frequency of all four H-bonds occurring simultaneously.*

patterns (see Fig 6). These results suggest that neither local hydrogen bonding nor global residue flexibility alone are predictors of catalytic performance, pointing to more complex structural or dynamic factors governing activity.

### Deep Learning based prediction of Enzyme Complex Structure cannot resolve differences between active and inactive variants

AlphaFold3 modeling of the enzyme-Roche ester and p-NPB complexes yielded nearly identical structures across all variants, with α-carbon RMSD values below 0.6 Å and active site backbone RMSD values under 0.18 Å. These minimal structural differences prevent differentiation between active and inactive enzyme states based solely on the predicted complexes. Fig 7 presents a superimposition of the AF3-predicted structures alongside the pairwise RMSD values for all investigated mutants. While all absolute RMSD values remain below 0.6 Å, most variants exhibit even higher structural similarity to one another. The largest deviations are observed when comparing mutant structures to wild-type LipA, particularly in complexes with either the Roche ester or p-NPB substrate.

We applied the recently published PLACER protocol, wherein a denoising network predicts the side chain and substrate conformations of protein-ligand interactions, previously used for engineering serine hydrolases [15], to analyze our LipA variants. However, the protocol failed to distinguish between active and inactive variants, as no significant differences in catalytic features were detected. Specifically, the predicted structures exhibited no correlation between enzymatic activity and the frequency of catalytic hydrogen bond formation (see Fig 8).

## Discussion

Our study successfully identified active LipA variants, demonstrating that even minor sequence changes can enhance catalytic activity. However, despite identifying functional mutants, we found no reliable way to reproducibly rank them using either our physics-based pipeline or state-of-the-art deep learning models. The observed activity differences—up to a tenfold increase—correspond to less than a 2 kcal/mol reduction in transition state energy, which is at the accuracy limit of DFT calculations. This suggests that the sensitivity of current computational methods may be insufficient to resolve the subtle energetic distinctions that govern enzymatic efficiency.

DFT simulations of the active site model revealed comparable energy barriers for transition states 1 and 3, indicating that both steps are rate-limiting. This finding underscores the importance of stabilizing multiple transition states in the design of multi-step enzymes such as serine hydrolases [15].

While deep learning has generated significant excitement in protein engineering, our results underscore critical limitations in applying these tools across different design targets. AlphaFold3 modeling of the enzyme-Roche ester complex produced nearly identical structures for all variants, with an α-carbon RMSD of <0.6 Å and an active site backbone RMSD of <0.18 Å. These minuscule structural differences do not allow distinguishing active and inactive enzyme states from static structure predictions alone. This aligns with recent findings that deep learning models primarily capture the most probable conformation, overlooking protein dynamic effects and transition states that are known to be essential for catalysis [59,60].

The inability of existing methods to resolve small activity differences suggests that current tools are better suited for identifying large structural shifts rather than refining single-point mutations. This limitation is particularly problematic in directed evolution, where activity improvements often occur through incremental optimizations [61]. Computational approaches like RFDiffusion or BayesDesign, which explore more extensive sequence and structural space, may be necessary to generate variants with sufficiently large activity shifts for reliable computational discrimination.

Our findings further highlight the urgent need for improved benchmarking of enzyme design pipelines. Existing structure-based approaches, including PLACER, failed to identify distinguishing features between active and inactive LipA variants, calling into question their applicability for subtle activity predictions [62]. While PLACER's ability to generate reaction intermediate ensembles for predicting preorganization remains unverified, its successful implementation as a step

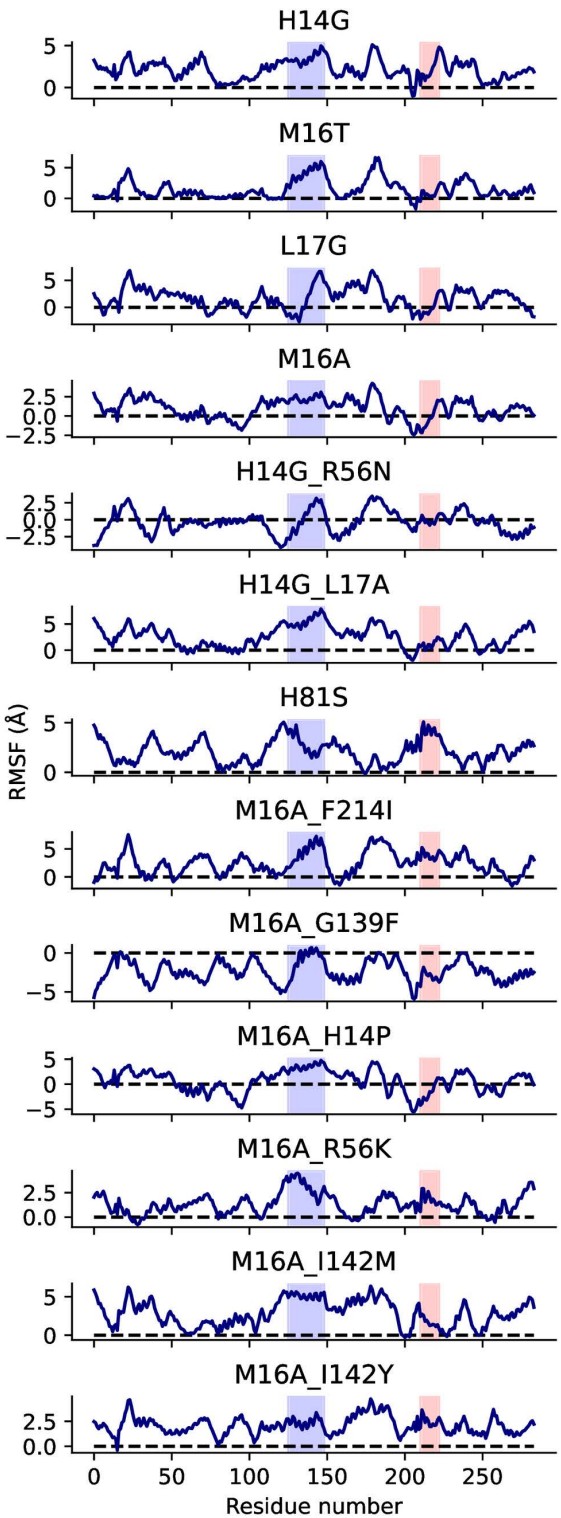

**Fig 6. Root Mean Square Fluctuations (RMSF) of all 13 tested LipA variants.** *The x-axis represents residue numbers, while the y-axis shows RMSF values in Å. Variants generally exhibit increased residue mobility compared to the wild-type enzyme, though no clear correlation with enzymatic activity is observed. Dotted lines indicate 0 Å RMSF.*

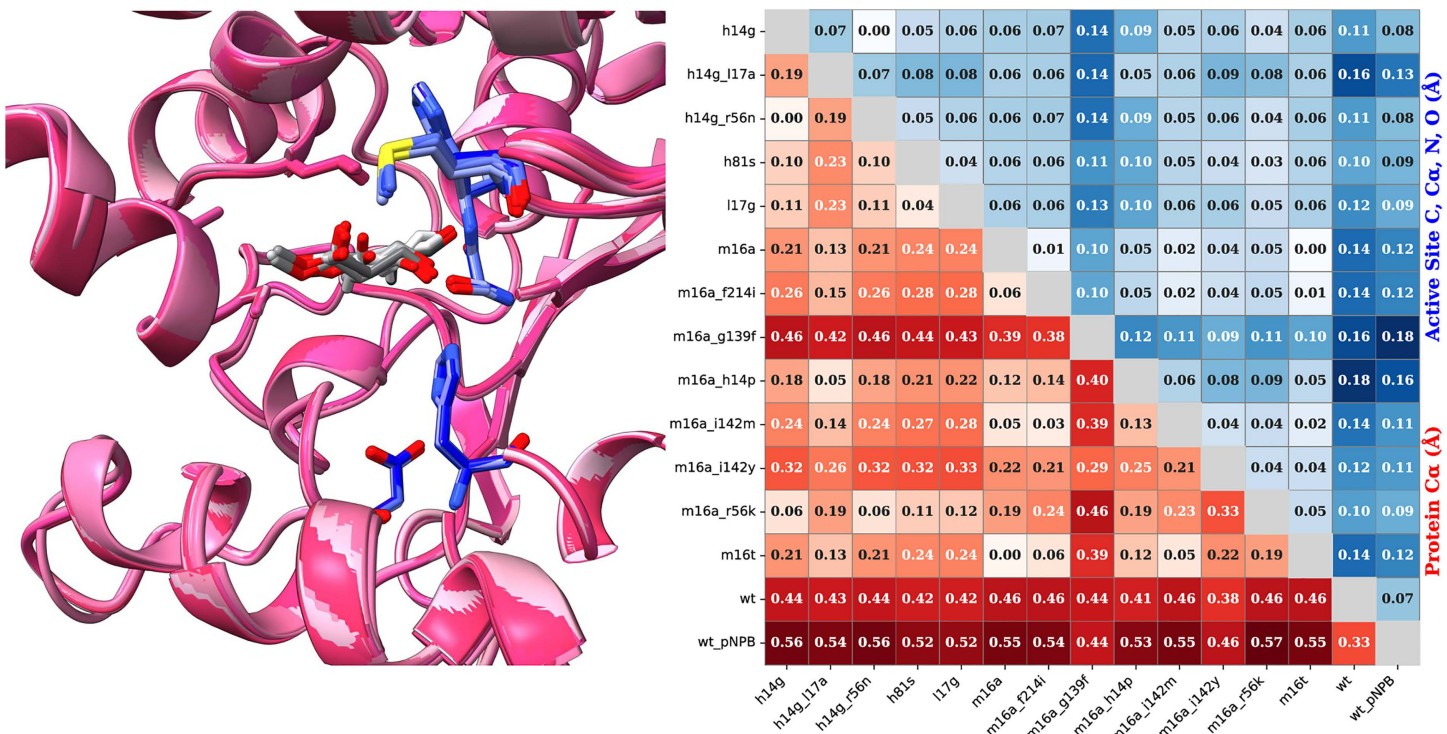

**Fig 7. AF3 predicted structural diversity of LipA variants.** *Left:The AF3 fold of all mutants with the (R)-Roche ester (backbone purple, active side residues blue, substrate grey). Right: Pairwise RMSD differences of AF3 structures of all mutants with the Roche ester and the wild-type enzyme with p-NPB – sorted by activity from left to right. The lower left red heatmap shows RMSD between the alpha carbons of variants. The upper right blue heatmap shows RMSD of the C, $C_\alpha$, N, and O atoms between active sites of variants (residues 16, 82, 83, 229, and 251).*

between analyzing static structural models and the dynamic catalytic landscapes could be critical for fine tuning enzymatic activities.

Ultimately, overcoming these limitations will require a paradigm shift in computational enzyme design. The accuracy of DFT calculations is insufficient to accurately capture the small energy differences in such complex systems that dictate enzymatic function. Future progress will likely depend on the integration of high-accuracy quantum chemical methods, such as coupled-cluster-based force fields, with deep learning-driven MD simulations capable of modeling fully atomistic, solvated environments. Advances in reactive force fields that can accurately capture transition state geometries, coupled with AI-driven methods that account for enzyme dynamics, will be essential to move beyond the current barriers. By developing more precise computational tools, we can begin to resolve the fine energetic and structural differences that define enzymatic efficiency, ultimately enabling the rational design of highly efficient biocatalysts.

## Methods

### Enzyme design protocol

We employed an ensemble-based computational design strategy to optimize the activation energy of the LipA-catalyzed reaction with the Roche ester substrate (see Fig 9). Starting from the LipA crystal structure (PDB: 1EX9), structural ensembles were generated using CONCOORD to sample conformational variability. Mutations were introduced via SCWRL4.0 [63], and both reactant [ES] and transition state [ES*] structures were modeled by adding back substrate geometries, with parameters obtained from DFT calculations (Fig 10A). The Gibbs free energy difference (ΔG)

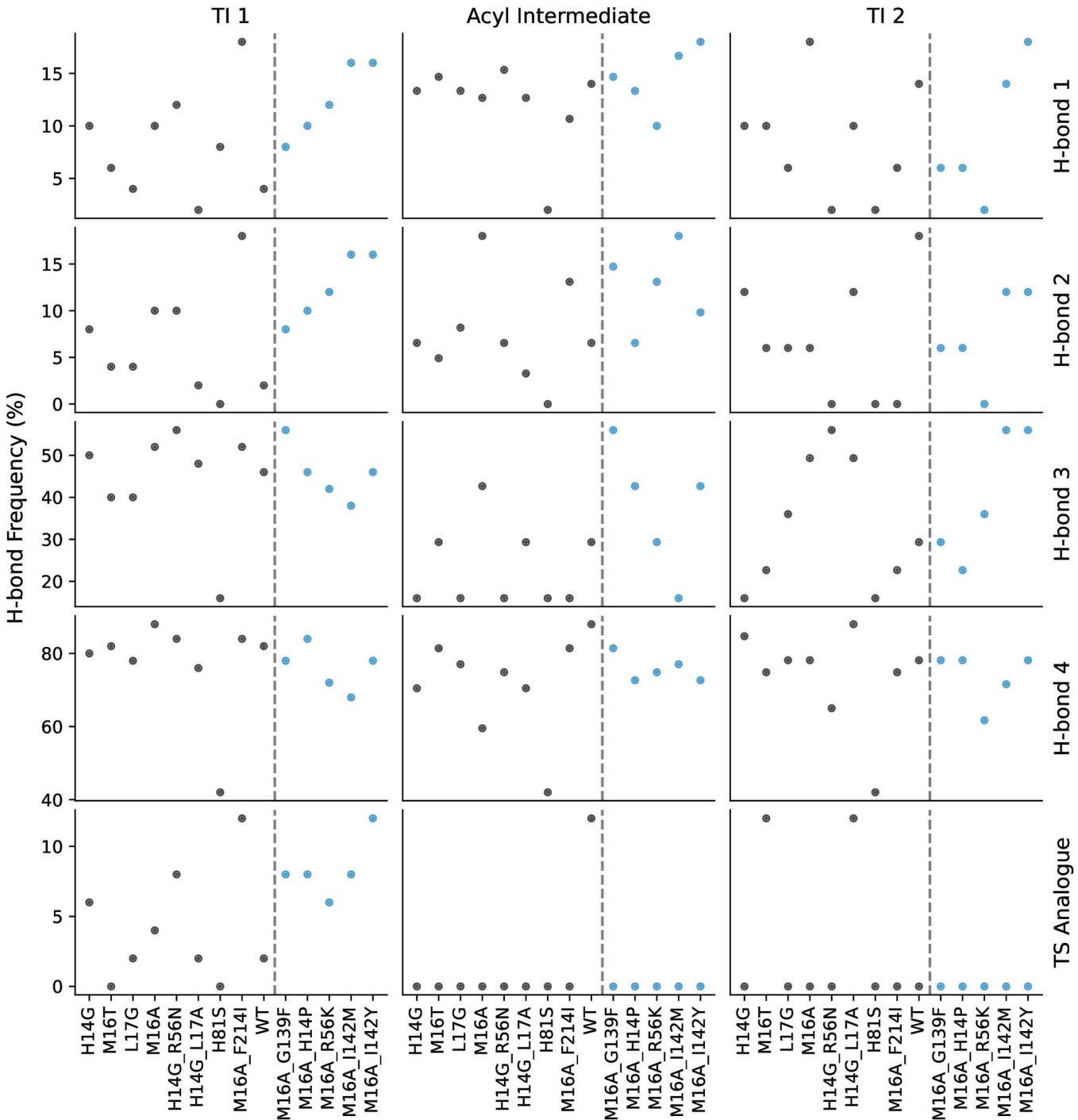

**Fig 8. The frequency of catalytic H-bond formation in PLACER output structures of active site catalyzing Roche ester hydrolysis.** *Each plot shows the average frequency of H-bond formation as a percentage (black and blue dots). The X-axis labels are the enzyme variants, sorted by activity high to low, left to right. Variants to the right of the dotted line (blue dots) showed 0 experimental activity towards the Roche ester. For each reaction intermediate (TI 1, Acyl Intermediate, TI 2), Ser82 was altered to match the corresponding structures in Fig 3. H-bonds 1 through 4 correspond to the H-bonds shown in Fig 4. The bottom row (Transition State analogue) shows the frequency of all four hydrogen bonds appearing in the same structure.*

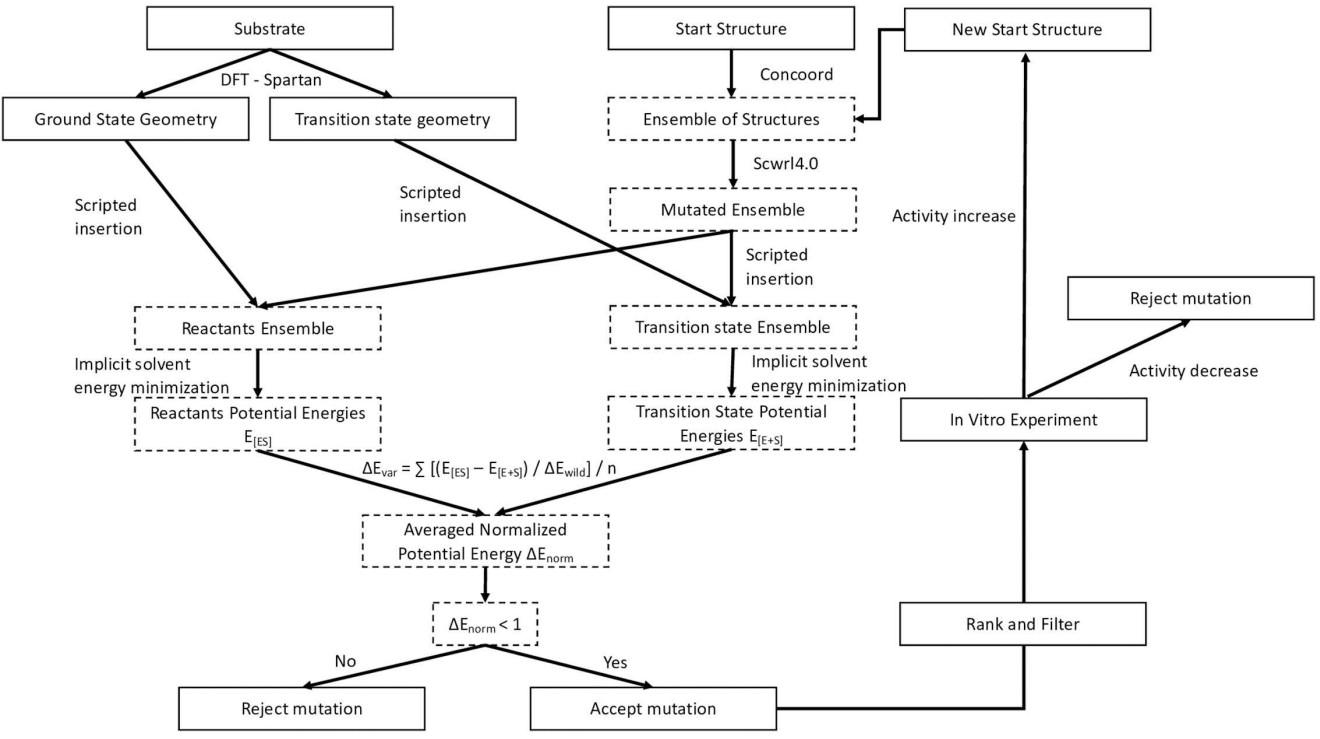

**Fig 9. Design protocol overview.** *Dashed boxes highlight elements of novelty in this protocol. For each mutation the residues in an ensemble were replaced and evaluated through energy minimization in ground state and tetrahedral conformation. Differences of calculated potential energies were normalized with respect to wild-type ensemble energy differences. Median of differences were reported as ΔE and used to classify variants into favorable or non-favorable candidates. For variants that yielded ΔE values below 1.0 and met other filter criteria, an in vitro experiment was used to verify improved activity. If an activity increase was measured the variant was accepted and served as new start structure.*

between the two states was approximated by the potential energy difference (ΔE), computed through energy minimization in implicit solvent using GROMACS with the AMBER99SB-ILDN forcefield (Figs 10B and 10C). Energy differences for mutated ensembles were normalized against the wild-type, and the median ΔE value (ΔEv) was used to rank mutations, mitigating the influence of outliers. Mutants with $ΔE_v < 1$ were considered promising candidates, with additional selection criteria applied to refine the selection. Experimental validation through lipase assays guided iterative rounds of design. For detailed protocols, including structure preparation, ensemble generation, mutation modeling, energy minimization, and PSSM calculations, refer to the Supplementary Sections S1–S5.

## Cloning, purification, activation and lipase activity of LipA mutants

The expression plasmids encoding five single-point mutants of LipA listed in Table 1 were created by PCR using Phusion DNA polymerase (Thermo Fischer Scientific) in whole plasmid amplification with mutagenic oligonucleotides (see S2 Table) designed for QuikChange method [64] and pLipA-SS [10] plasmid as a template. Eight double-point mutants were generated the same as the single-point mutants, repeating the mutation procedure using a plasmid that already contained a single-point mutation. Production and purification of LipA wild-type and mutants were performed by following the procedure described by Hausmann et al. [56] Purity of LipA preparations was judged by sodium dodecyl sulfate-polyacrylamide gel electrophoresis (SDS-PAGE) under denaturation conditions on 16% (w/v) gels followed by staining with Coomassie Brilliant Blue G250. [65] *In vitro* activation of LipA with specific chaperon was conducted over night at 4°C, to ensure

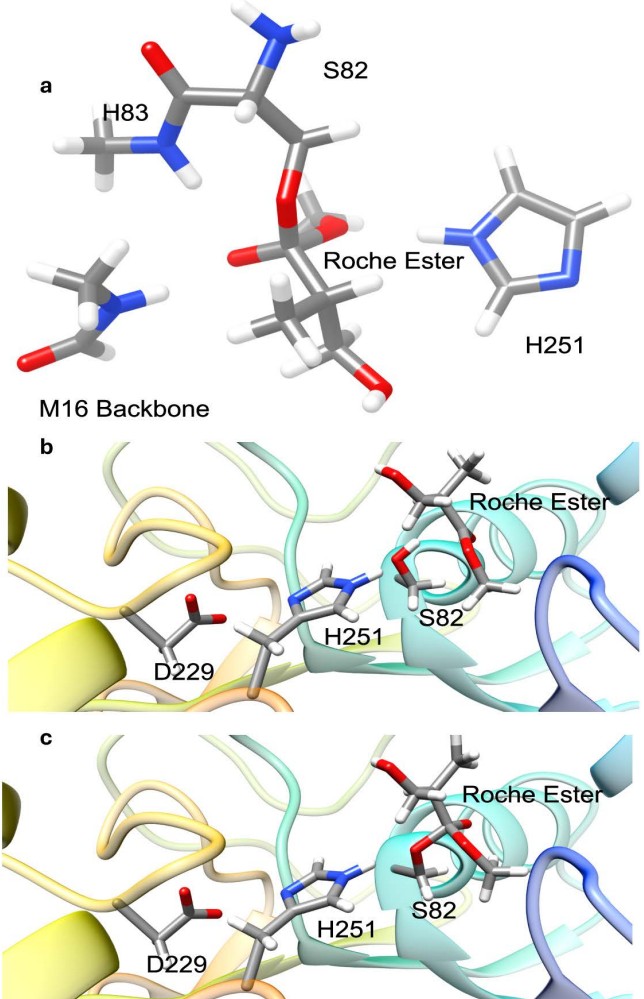

**Fig 10. LipA active site models.** *a) Active site model and Roche ester configuration as used for force field parametrization (see Methods for details on DFT calculation). b) The ground state Roche ester docked but not covalently bound to LipA (colored cartoon). This represents the reactant state ES. In addition, the sidechains of the catalytic triad are shown. c) The transition state geometry of Roche ester covalently bound to S82 of LipA (colored cartoon). This tetrahedral intermediate represents the ES\* state, again the catalytic triad is shown.*

complete activation, according to Hausmann et al. [56]. Lipase activity of activated LipA was measured with *para*-nitrophenyl butyrate (*p*-NPB). [66]

## Enzymatic hydrolysis of Roche ester

The activity of activated wild-type, single- and double-point mutants of LipA (3.6 μM) towards a racemic mixture of Roche ester (2 mM) was determined by gas chromatographic analysis.

The LipA (3.6 μM) in Tris-HCl buffer (50 mM, pH 8, 3.5 mM $CaCl_2$, 0.7 mM lauryl maltoside, 45% (v/v) glycerol) was incubated at 30°C with Roche ester (2 mM) and samples were taken after 2, 4, 6, 23.5, 27.5 and 74.5 h. Reaction mixtures (25 μL) were extracted with double amount of methyl-tert-butylether and residual Roche ester was quantified with gas chromatograph GC 17A (Shimadzu, Kyoto, Japan) equipped with flame ionization detector using CP Chirasil DEX CB column and helium flow of 1.3 mL $min^{-1}$. The separation was performed using the following program: 5 min, 60°C followed

by increasing the temperature for 5°C/min during 27 min. Enzyme activity was expressed as decrement of the area under the signal's characteristic peak for R-Roche ester (at 15.467 min) and S-Roche ester (at 14.783 min) with time.

### DFT characterization of ester hydrolysis mechanism

M06-2X/6-31G(d,p) DFT calculations were performed using Gaussian 16 [67]. The M06-2X functional is a hybrid meta-GGA method incorporating 54% Hartree–Fock exchange, developed for accurate main-group thermochemistry, kinetics, and noncovalent interactions. The 6-31G(d,p) split-valence double-zeta basis set includes polarization functions (d on heavy atoms, p on hydrogens), which improve the description of electron distribution changes during bond formation and cleavage in the enzymatic reaction. The active site model was derived from the crystal structure (PDB: 1EX9), which includes a covalently bound substrate analog (1,2-dioctylcarbamoyl-glycero-3-O-p-nitrophenyl octylphosphonate) that closely mimics the transition state conformation of the lipid hydrolysis addition step. The model encompassed the side chains of SER 82, HIS 251, ASP 229, and the backbone amines of MET 16 and HIS 83.

We manually capped the termini with methyl groups, and the substrate analog was replaced with either the (R)- or (S)-Roche ester. The theozyme was adjusted for each hydrolysis step: the base model was used for the addition step, the methoxy-carbonyl bond of the ester-serine complex was manually elongated for methanol elimination, the methoxy group and serine hydroxyl hydrogen were replaced with a water molecule in proximity to the ester's carbonyl carbon for hydroxide addition, and the methoxy group was substituted with a hydroxide for acid elimination [68]. While the crystal structure was used as the initial conformation for each transition state model, no constraints were applied to fix the positions of amino acid atoms. No implicit or explicit solvent model was applied; all DFT calculations were performed in the gas phase.

To explore alternative active site conformations, CREST and GFN2-xTB was employed to generate conformer ensembles. Two constraint strategies were applied: (1) freezing all amino acid atoms while constraining the ester carbonyl carbon–serine hydroxyl oxygen distance, or (2) constraining the ester/serine distance along with the α- or β-carbons of each amino acid fragment. Additional distance constraints were applied to heavy atom pairs involved in hydrogen bonds. Up to 20 distinct conformers from each ensemble were selected and screened for viable transition states using DFT calculations, which were ran without any position or bond length constraints.

This calculation pipeline was executed for both the (R)- and (S)-Roche esters, including full reactant and product optimizations for each identified transition state.

### Non-standard amino acid parameterization for transition state MD simulations

To model transition state conformations in MD simulations, non-standard amino acid parameters were developed. For the ester addition step, SER 82 was replaced with a deprotonated serine (see S3 Fig). For the methanol and hydroxide elimination steps, SER 82 was substituted with a deprotonated serine covalently bound to the ester carbonyl group (see S4 Fig).

The parameterization pipeline involved manual construction of the modified residue, followed by conformer generation with CREST (freezing backbone dihedrals at α- or β-carbons). Conformers were optimized in Gaussian 16 (M06-2X/6-31G(d,p)), and AMBER-compatible partial charges were computed (HF/6-31G). AmberTools 21 parameterized the residue in the GAFF2 force field, producing.prep files [69]. An in-house python script (made available for download as Supplementary Material) assigned GAFF-2/ff14SB interface parameters, saved in.frcmod files. AmberTools then generated solvated system topology and coordinate files, integrating the non-standard residue for MD simulations.

### Transition state MD simulations and analysis

Initial MD simulations of LipA and its mutations were conducted using the AMBER99SB-ILDN force field within GROMACS [70]. AMBER99SB-ILDN is a widely validated protein force field incorporating refined side-chain torsion parameters for isoleucine, leucine, aspartate, and asparagine residues. These adjustments improve the accuracy of side-chain dynamics,

which can enhance modeling of hydrophobic contacts and packing, such as those critical in lipase active sites. The LipA crystal structure (PDB code 1EX9) was stripped of solvent and substrate molecules, energy minimized in a vacuum and solvated with explicit TIP3P water. After neutralizing charges with ions, the system underwent equilibration for 100 ps in NVT and NPT ensembles at 300 K, followed by a 100 ns production run. Mutants P123E and K223G were generated with SCWRL4.0 [63], and the simulation protocol was repeated for each mutant.

Subsequent MD simulations were conducted with OpenMM and the AMBER14-ffSB force field to model the transition states identified with DFT calculations [71,72]. The crystal structure was processed by removing solvent molecules, adding missing atoms, and replacing the substrate analog with the (R)-Roche ester, hydroxide ion, or methoxide ion. Mutations at SER 82 were introduced using AmberTools, and sodium ions neutralized formal charges. A bound calcium ion was substituted with harmonic potentials (k = 1,254,000 kJ/mol/nm²). The system was solvated in a 10Å TIP3P water box, equilibrated at 300 K for 5 ps, and simulated for 100–150 ns with a Langevin integrator, repeated three times per mutant and transition state.

In the first round, a harmonic potential constrained the substrate to model bond formation or cleavage, with bond lengths set to DFT averages (e.g., 1.9 Å for TS 1, 1.93 Å for TS 2/3). A second simulation round, without this constraint, was performed for TS 1, running 100 ns.

Trajectory analysis focused on key hydrogen bonds observed in DFT simulations: (1) ester carbonyl oxygen–residue 16 amine, (2) ester carbonyl oxygen–residue 83 amine, (3) SER 82 hydroxy oxygen–HIS 251 imidazole nitrogen, and (4) HIS 251 pyrrole nitrogen–ASP 229 carboxyl oxygen. Bond frequencies for each mutant and reaction step are shown in Fig 5, informing mutation impact on transition state stabilization.

### AlphaFold3 analysis

AlphaFold3 structures were generated by running AF3 locally, maintaining consistency across runs by using the same three random seeds for each structure. Ligands, represented as SMILES strings, were incorporated into the input files following the guidelines provided in the official GitHub documentation.

For structural comparisons, the highest-ranked model for each LipA mutant was aligned using ChimeraX [73]. Pairwise root-mean-square deviation (RMSD) values were computed for α-carbon atoms and for the backbone atoms of the active site residues using the Biopython package [74]. This allowed for a precise evaluation of structural differences between variants at both the global and active site levels.

### PLACER analysis

PLACER structures were generated following the methodology described by Lauko et al. [15]. Active site models were constructed to represent key catalytic intermediates, including Tetrahedral Intermediates 1 and 2, as well as the Acyl Intermediate (see Fig 3). Modified serine structures were created using GaussView 6 [75] and ChimeraX [73] to accurately capture the geometric changes associated with catalysis.

For each LipA mutant and intermediate state, 50 active site models were generated to account for structural variability. The resulting structures were processed in ChimeraX, and catalytic hydrogen bond formation was assessed using a custom Python script leveraging the MDAnalysis package [76]. This approach allowed for a systematic evaluation of hydrogen bond patterns and their potential correlation with enzymatic activity.

### Supporting information

**S1 File. Preparation of the LipA and substrate structure; S2: Generation of CONCOORD ensemble; S3: Modeling of mutations; S4: Energy minimizations; S5: Calculation of the PSSM; S6: Selection and ranking of mutations.**
(DOCX)

## Author contributions

**Conceptualization:** Karl-Erich Jaeger, Gunnar F. Schröder, Dennis Della Corte.

**Data curation:** Dennis Della Corte.

**Formal analysis:** Spencer Gardiner, Filip Kovacic, Dennis Della Corte.

**Funding acquisition:** Karl-Erich Jaeger, Gunnar F. Schröder, Dennis Della Corte.

**Investigation:** Spencer Gardiner, Peter Dollinger, Filip Kovacic, Jörg Pietruszka, Daniel H. Ess, Dennis Della Corte.

**Methodology:** Daniel H. Ess, Karl-Erich Jaeger, Gunnar F. Schröder, Dennis Della Corte.

**Project administration:** Filip Kovacic, Karl-Erich Jaeger, Gunnar F. Schröder, Dennis Della Corte.

**Resources:** Jörg Pietruszka, Karl-Erich Jaeger, Gunnar F. Schröder, Dennis Della Corte.

**Software:** Spencer Gardiner, Dennis Della Corte.

**Supervision:** Filip Kovacic, Gunnar F. Schröder, Dennis Della Corte.

**Validation:** Spencer Gardiner, Filip Kovacic, Daniel H. Ess, Karl-Erich Jaeger, Dennis Della Corte.

**Visualization:** Spencer Gardiner, Dennis Della Corte.

**Writing – original draft:** Spencer Gardiner, Dennis Della Corte.

**Writing – review & editing:** Spencer Gardiner, Peter Dollinger, Filip Kovacic, Jörg Pietruszka, Daniel H. Ess, Karl-Erich Jaeger, Gunnar F. Schröder, Dennis Della Corte.

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
