## [Decision Letter · Decision Letter 0]

31 Jul 2025

PONE-D-25-23633Resolution of physics and deep learning-based protein engineering filters : A case study with a lipasePLOS ONE

Dear Dr. Della Corte,

Thank you for submitting your manuscript to PLOS ONE. After careful consideration, we feel that it has merit but does not fully meet PLOS ONE’s publication criteria as it currently stands. Therefore, we invite you to submit a revised version of the manuscript that addresses the points raised during the review process. Please submit your revised manuscript by Sep 14 2025 11:59PM. If you will need more time than this to complete your revisions, please reply to this message or contact the journal office at plosone@plos.org . Please include the following items when submitting your revised manuscript:

We look forward to receiving your revised manuscript.

Kind regards,

Zeynep Ozdemir

Academic Editor

PLOS ONE

Journal Requirements:

D.DC and S.G. were supported by the National Institute of General Medical Sciences of the National Institutes of Health under award number R15GM155803. D.DC. and P.D. thank the graduate school iGRASPseed for funding. Part of this study was supported by the Deutsche Forschungsgemeinschaft (DFG, German Research Foundation) through funding no. JA 448/8-1 to KEJ. G.F.S. and D.DC.

D.DC and S.G. were supported by the National Institute of General Medical Sciences of the National Institutes of Health under award number R15GM155803. D.DC. and P.D. thank the graduate school

22 iGRASPseed for funding. Part of this study was supported by the Deutsche Forschungsgemeinschaft (DFG, German Research Foundation) through funding no. JA 448/8-1 to KEJ. G.F.S. and D.DC. gratefully acknowledge the computing time granted by the JARA-HPC Vergabegremium and VSR commission on the supercomputer JUROPA at Forschungszentrum Jülich.

D.DC and S.G. were supported by the National Institute of General Medical Sciences of the National Institutes of Health under award number R15GM155803. D.DC. and P.D. thank the graduate school iGRASPseed for funding. Part of this study was supported by the Deutsche Forschungsgemeinschaft (DFG, German Research Foundation) through funding no. JA 448/8-1 to KEJ. G.F.S. and D.DC.

4. Please remove all personal information, ensure that the data shared are in accordance with participant consent, and re-upload a fully anonymized data set.

Additional guidance on preparing raw data for publication can be found in our Data Policy (https://journals.plos.org/plosone/s/data-availability#loc-human-research-participant-data-and-other-sensitive-data) and in the following article: http://www.bmj.com/content/340/bmj.c181.long .

Additional Editor Comments:

Thank you very much for submitting your manuscript to PLOS One. Your manuscript has now been reviewed by experts in the field. Please revise the manuscript according to the referees' comments and upload the revised file.

Reviewers' comments:

Reviewer's Responses to Questions

**Comments to the Author**

1. Is the manuscript technically sound, and do the data support the conclusions?

Reviewer #1: Yes

2. Has the statistical analysis been performed appropriately and rigorously? 

Reviewer #1: Yes

3. Have the authors made all data underlying the findings in their manuscript fully available?

Reviewer #1: Yes

4. Is the manuscript presented in an intelligible fashion and written in standard English?

Reviewer #1: No

5. Review Comments to the Author

Reviewer #1: This study focuses on evaluating the limitations of enzyme design methods through computation in optimizing enzyme activity against non-natural substrates, specifically using the LipA enzyme from Pseudomonas aeruginosa to hydrolyse the Roche ester substrate. There are some of my inputs that can improve the quality of this article.

1) The title is very general. please be more specific

2) Please specify the selection of H14/M16/L17/H81!

3) Compare the RMSF (or RMSD relative to the crystal structure) in the CONCOORD ensemble vs. MD 100 ns. In this article, CONCOORD is mentioned to “lower the pressure” due to constraints. However, doesn't this make the ensemble too “too constrained”, thus losing millisecond-scale motions that can affect binding?

4) The author does not provide an explanation of the computational methods used, for example, AMBER99SB-ILDN and M06-2X/6-31G. Please clarify!

5) Please specify the solvent conditions in the DFT calculations!

6) In Figure 5, what specific criteria (distance & angle) were used to detect H-bonds in the analysis? Were other thresholds (eg: 3.2 Å) tested to see if the pattern changed?

6. PLOS authors have the option to publish the peer review history of their article (what does this mean? ). If published, this will include your full peer review and any attached files.

**Do you want your identity to be public for this peer review?** For information about this choice, including consent withdrawal, please see our Privacy Policy .

Reviewer #1: No

---

## [Author Response · Author response to Decision Letter 1]

13 Aug 2025

We uploaded a point by point response, including responses to the editors and reviewers requests.

---

## [Decision Letter · Decision Letter 1]

31 Aug 2025

Resolution of physics and deep learning-based protein engineering filters : A case study with a lipase for industrial substrate hydrolysis

PONE-D-25-23633R1

Dear Dr. Della Corte,

We’re pleased to inform you that your manuscript has been judged scientifically suitable for publication and will be formally accepted for publication once it meets all outstanding technical requirements.

Kind regards,

Zeynep Ozdemir

Academic Editor

PLOS ONE

Additional Editor Comments (optional):

Reviewer #1: All comments have been addressed.

Reviewers' comments:

Reviewer's Responses to Questions

**Comments to the Author**

1. If the authors have adequately addressed your comments raised in a previous round of review and you feel that this manuscript is now acceptable for publication, you may indicate that here to bypass the “Comments to the Author” section, enter your conflict of interest statement in the “Confidential to Editor” section, and submit your "Accept" recommendation.

Reviewer #1: All comments have been addressed

2. Is the manuscript technically sound, and do the data support the conclusions?

Reviewer #1: Yes

3. Has the statistical analysis been performed appropriately and rigorously? 

Reviewer #1: Yes

4. Have the authors made all data underlying the findings in their manuscript fully available?

Reviewer #1: Yes

5. Is the manuscript presented in an intelligible fashion and written in standard English?

Reviewer #1: Yes

6. Review Comments to the Author

Reviewer #1: The revisions provided can be considered satisfactory. The authors have:

1) Addressed scientific transparency: They are now more open in explaining their methodological approach, assumptions, and limitations, so readers understand the context of the results presented.

2) Provided sufficient definition of controversies: Discrepancies between simulation and experimental results are no longer left "hanging," but rather the causes or possible underlying factors are explained.

3) Acknowledged study limitations: The discussion section now emphasizes the limitations of the computational method (for instance, DFT accuracy, parameterization sensitivity), so the conclusions do not seem too absolute.

4) Avoided overinterpretation: The results are now read more cautiously, emphasizing that the substrate recognition mechanism is influenced not only by cargo distribution at the terminal, but also by other, more complex factors.

With these revisions, it is clear that the authors accommodated reviewer's comments and improved the quality of the manuscript. I am satisfied, and the manuscript is worthy of further consideration for publication.

7. PLOS authors have the option to publish the peer review history of their article (what does this mean? ). If published, this will include your full peer review and any attached files.

**Do you want your identity to be public for this peer review?** For information about this choice, including consent withdrawal, please see our Privacy Policy .

Reviewer #1: No

---

## [Editor Report · Acceptance letter]

PONE-D-25-23633R1

PLOS ONE

Dear Dr. Della Corte,

I'm pleased to inform you that your manuscript has been deemed suitable for publication in PLOS ONE. Congratulations! Your manuscript is now being handed over to our production team.

Kind regards,

on behalf of

Professor Zeynep Ozdemir

Academic Editor

PLOS ONE